# Coordinate Transform Fourier Neural Operators for Symmetries in Physical Modelings

**Wenhan Gao**  *wenhan.gao@stonybrook.edu*
*Stony Brook University*

**Ruichen Xu**  *ruichen.xu@stonybrook.edu*
*Stony Brook University*

**Hong Wang***  *wanghong1700@gmail.com*
*Stony Brook University*

**Yi Liu**  *yi.liu.4@stonybrook.edu*
*Stony Brook University*

**Reviewed on OpenReview:** *https://openreview.net/forum?id=pMD7A77k3i*

## Abstract

Symmetries often arise in many natural sciences; rather than relying on data augmentation or regularization for learning these symmetries, incorporating these inherent symmetries directly into the neural network architecture simplifies the learning process and enhances model performance. The laws of physics, including partial differential equations (PDEs), remain unchanged regardless of the coordinate system employed to depict them, and symmetries sometimes can be natural to illuminate in other coordinate systems. Moreover, symmetries often are associated with the underlying domain shapes. In this work, we consider physical modelings with neural operators (NOs), and we propose an approach based on coordinate transforms (CT) to work on different domain shapes and symmetries. Canonical coordinate transforms are applied to convert both the domain shape and symmetries. For example, a sphere can be naturally converted to a square with periodicities across its edges. The resulting CT-FNO scheme barely increases computational complexity and can be applied to different domain shapes while respecting the symmetries. The code and data are publicly available at `https://github.com/wenhangao21/CTFNO`.

## 1 Introduction

In numerous fields, such as electromagnetism (Bermúdez et al., 2014), researchers seek to study the behavior of physical systems under various parameters, such as different initial conditions, boundary values, and forcing functions. Traditional numerical methods can be excessively time-consuming for simulating such physical systems. A class of data-driven surrogate models, termed neural operators (NOs), provide an efficient alternative (Li et al., 2021; Rahman et al., 2023; Raonic et al., 2023; Chen et al., 2023). Neural operators approximate the mapping from parameter function space to solution function space. Once trained, obtaining a solution for a new instance of the parameter requires only a forward pass of the network, which can be several orders of magnitude faster than traditional numerical methods (Li et al., 2021). A prevalent field of application is solving classes of Partial differential equations (PDEs). PDEs are widely used in modeling physical phenomena; for example, Navier-Stokes equations in fluid dynamics (Serrano et al., 2023) and Schrödinger equations (Dirac & M., 1981) in quantum mechanics. Although solving parametric PDEs is a major application of neural operators, they have wider applicability in general physical modeling, even without known physics formulations, such as in climate modeling (Bonev et al., 2023).

---

*Work done during an internship at Stony Brook University.

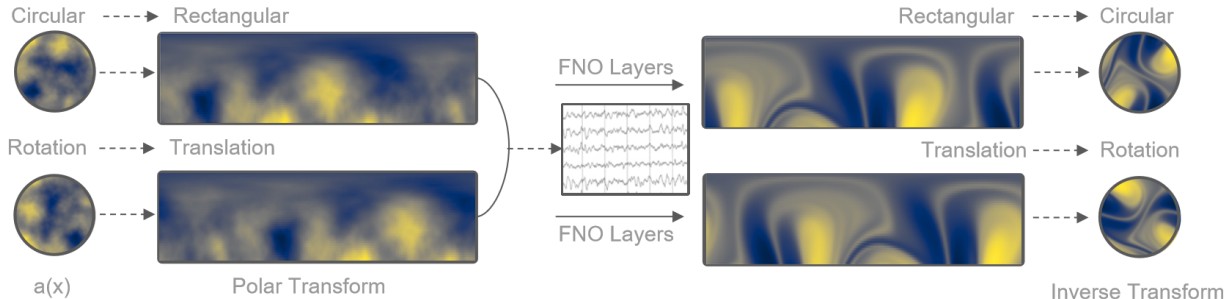

Figure 1: Overview of CT-FNO. We first apply an appropriate coordinate transform so that the domain becomes rectangular, and the corresponding symmetry becomes translation symmetry. Then, FNO layers, which are translationally equivariant, operate in a different coordinate system to predict output solutions. Finally, an inverse coordinate transform is applied back to the original coordinate system.

Various symmetries can be naturally observed in the dynamics of physical phenomena. Symmetries within the underlying physical phenomena are reflected both in the physical model and its solution operators. Symmetries in physical systems can manifest in various ways, indicating specific properties within the mathematical formulation of the physical systems. For instance, in climate modeling, we observe rotational equivariance as the Earth rotates. In PDEs, if a PDE describing a physical phenomenon remains unchanged when the spatial coordinates or time are shifted, it displays translational symmetry. Common symmetries include translation symmetries, rotation symmetries, scale symmetries, Galilean symmetries, and more. Incorporating these symmetries into network architectures can enhance model generalization, interpretability, and reduce learning complexity (Cohen & Welling, 2016; 2017; Weiler et al., 2023; Zhang et al., 2023).

Several group equivariant architectures have been proposed and explored in diverse applications to incorporate symmtries (Kofinas et al., 2021; Weiler & Cesa, 2019; Cohen et al., 2019; Bekkers et al., 2018; Fuchs et al., 2020). Group convolutional neural networks have achieved significant success in numerous symmetry groups, encompassing Euclidean groups $E(n)$, scaling groups, and others. Nevertheless, group convolutions can be computationally expensive for general continuous groups (Cohen & Welling, 2017), and for some symmetry classes, particularly those arising in PDEs, performing group convolutions can even be intractable.

In this work, we introduce the Coordinate Transform Fourier Neural Operator (CT-FNO) to address symmetries within physical modelings. Employing a canonical coordinate transform, we demonstrate how different symmetries can be seamlessly transformed into translation symmetries. Leveraging the properties of the convolution, Fourier layers in FNO exhibit equivariance to cyclic translations. Consequently, the CT-FNO proposed herein can effectively harness symmetries, offer generalization across various domain shapes, and does not introduce any new learnable parameters. **Our contribution can be summarized in four parts:** (1) We propose the CT-FNO framework, which extends FNO into various domains while respecting the underlying symmetries. (2) It is mathematically justified that CT-FNO is capable of approximating the behavior of an operator within a specified precision, and this lays the foundation for our CT-FNO framework. (3) Experimental results reveal that CT-FNO can effectively maintain symmetries. In instances where symmetries exist, CT-FNO consistently outperforms FNO with a minimum increase in computation cost. (4) Experimental results also demonstrates generalization applicability of CT-FNO across different domain shapes while respecting the underlying symmetries, which is not achievable by existing architectures.

## 2 Related Work

### 2.1 Neural PDE Solvers

Recently, neural PDE solvers have shown great success as an alternative to traditional numerical methods for solving PDE problems in many areas of practical engineering and life sciences (Sirignano et al., 2020; Pathak et al., 2022; Zhang et al., 2022; Azizzadenesheli et al., 2024). Traditionally, solving a PDE involves seeking a smooth function satisfying the derivative relationships within the equations. Based on this view, Physics-Informed Neural Networks (PINNs) have been developed to tackle PDEs individually. Another perspective

is to regard differential operators as mappings between function spaces. Stemming from this perspective, neural operators have been proposed for solving families of PDEs (Anandkumar et al., 2019; Lu et al., 2021; Brandstetter et al., 2022). Neural operators parameterize solution operators in infinite dimensions and map the input parameter functions to their respective solutions. Neural operators can be applied beyond PDEs; they can serve as surrogates for integral operators, derivative operators, or general function-to-function mappings (Fanaskov & Oseledets, 2022). Among neural operators, Fourier neural operators (FNOs) (Li et al., 2021) have drawn lots of attention. FNO is a type of kernel integral solver in which the kernel integral is imposed to be a convolution (Kovachki et al., 2023). Global convolution is performed via Fourier layers in the frequency domain. By leveraging low-frequency modes and truncating high-frequency modes, FNO captures global information and incurs low computational costs and achieves discretization invariance.

## 2.2 Equivariant Architectures

The concept of equivariance emerge in many machine learning tasks, especially in the field of computer vision. Convolutional Neural Networks (CNNs) are known to be equivariant to translations, a characteristic that has propelled CNNs' significant success. Group convolution has been studied to achieve equivariance beyond translation (Cohen & Welling, 2016), such as rotation and scaling. A regular group convolution, within a discrete group, convolves input features with multi-channel filters in the group space; a group action on the input feature corresponds to cyclic shifts between channels. Later works extend equivariance to continuous groups, for example, utilizing steerable filters (Cohen & Welling, 2017; Weiler & Cesa, 2019) and B-spline interpolation (Bekkers, 2020). Other than group convolution, Esteves et al. (2018) brings up an interesting idea of achieving rotation and scale equivariance by using a log-polar sampling grid. For symmetries in PDEs, Group FNO (G-FNO) (Helwig et al., 2023) parameterized convolution kernels in the Fourier-transformed group space, extending group equivariant convolutions to the frequency domain. The resulting G-FNO architecture is equivariant to rotations, reflections, and translations. Nevertheless, group convolutions can be computationally expensive; for some symmetries arising in PDEs or physical dynamics in general, performing group convolutions can even be intractable. Symmetries such as rotation and translation are commonly studied in computer vision. However, those that exist in PDEs or physical dynamics, in general, are under-studied. Our work, instead of utilizing group convolution, offers an alternative approach to incorporating symmetries in neural operators through canonical coordinate transformation.

## 2.3 Relations with Prior Work

G-FNO (Helwig et al., 2023) is a recent method that performs group convolution in the frequency domain. As the Fourier transform is factually rotationally equivariant, G-FNO also achieves equivariance in the physical domain. However, in both FNO and G-FNO, in order to benefit from the power of FFT, the domain has to be rectangular, which limits the symmetries it can have. For example, with rotation symmetries, a rectangular domain cannot be rotationally equivariant to arbitrary rotations. Unlike in computer vision tasks, where the convolution kernels are usually local and can have local symmetries, the kernel in FNO and G-FNO is global. Thus, the existence of local symmetries is unclear. For PDEs, the symmetries they can have often associate with the underlying domain shapes. Therefore, we propose CT-FNO to adapt to various domain shapes while also respecting the symmetries that the underlying domain shape can embody. Although both our work and G-FNO aims to incorporate symmetries, we look at a different angle that takes both symmetries and domain shapes into account, which cannot be solved solely by group convolution.

# 3 Preliminaries

## 3.1 Group Equivariance

Equivariance is a property of an operator, such as a neural network or a layer within one, such that if the input transforms, the output transforms correspondingly in a predictable way. A more complete review of necessary group theory and equivariance concepts within the scope of this work is included in Appendix A.

**Definition 3.1** (Equivariance). An operator $\Phi : \mathcal{X} \to \mathcal{Y}$ is said to be equivariant to a group $G$ if actions of $G$ on $\mathcal{X}$ and $\mathcal{Y}$, respectively denoted by $L_g^{\mathcal{X}} : \mathcal{X} \to \mathcal{X}$ and $L_g^{\mathcal{Y}} : \mathcal{Y} \to \mathcal{Y}$, satisfy

$$\forall g \in G : \; L_g^{\mathcal{Y}}(\Phi(f)) = \Phi(L_g^{\mathcal{X}} f), \; f \in \mathcal{X},$$

where $\mathcal{X}$ and $\mathcal{Y}$ are subspaces of a function space. In other words, the operator $\Phi$ commutes with actions of $G$.

In the context of operator learning, $L_g^{\mathcal{X}}$ and $L_g^{\mathcal{Y}}$ can be considered as transformations, such as rotation, of the input parameter function and the output solution function, respectively. $\Phi$, on the other hand, can be thought of as the solution operator that maps an input function to its corresponding output solution. Equivariance to various groups, including but not limited to $SE(n), E(n), \mathrm{SIM}(n)$, can be achieved by group-convolutions (Cohen & Welling, 2016; Weiler et al., 2023). Let $v(h)$ and $\kappa(h)$ be real valued functions on group $G$ with $L_g v(h) = v\left(g^{-1}h\right)$, the group convolution is defined as:

$$(v \star \kappa)(h) = \int_{g \in G} v(g)\kappa\left(g^{-1}h\right) dg,$$

where $v$ can be regarded as the input feature and $\kappa$ the convolution kernel.

Note that the first group convolution layer should be treated differently if the input functions are not defined on $G$. For semi-direct product groups, $\mathbb{R}^d \rtimes H$, the first layer is a lifting convolution that raises the feature map dimensions to the group space (Bekkers, 2020).

Integrability over a group and the identification of the suitable measure, $dg$, are necessary for group convolution. It has been shown that with the measure $dg$, group convolution consistently maintains group equivariance. For all $a \in G$,

$$
\begin{aligned}
(L_a v \star \kappa)(h) &= \int_{g \in G} v\left(a^{-1}g\right) \kappa\left(g^{-1}h\right) dg \\
&= \int_{b \in G} v(b)\kappa\left((ab)^{-1}h\right) db \\
&= \int_{b \in G} v(b)\kappa\left(b^{-1}a^{-1}h\right) db \\
&= (v \star \kappa)\left(a^{-1}h\right) \\
&= L_a(v \star \kappa)(h).
\end{aligned}
$$

Similar arguments can show that the first layer, sometimes referred to as the lifting convolution layer, also upholds group equivariance.

### 3.2 Fourier Neural Operators

Fourier neural operators learns to map an input function to the solution function in infinite-dimensional spaces. Inspired by the kernel method for PDEs, each Fourier layer consists of a fixed non-linearity and a kernel integral operator $\mathcal{K}$ modeled by network parameters, defined as $(\mathcal{K}v)(x) = \int \kappa(x,y)v(y)\mathrm{d}y$. As a special case of integral kernel operator, translation invariance is imposed on the kernel, $(\kappa(x,y) = \kappa(x-y))$, which is a natural choice from the perspective of fundamental solutions. Thus, the integral kernel operator in FNO is defined as a convolution operator

$$(\mathcal{K}v)(x) = \int_{\mathbb{R}^d} \kappa(x-y)v(y)dy. \tag{1}$$

From the famous convolution theorem, convolution in physical space can be carried out as element-wise multiplication in the frequency domain,

$$(\mathcal{K}v)(x) = \mathcal{F}^{-1}(\mathcal{F}\kappa \cdot \mathcal{F}v)(x), \tag{2}$$

where $\mathcal{F}$ and $\mathcal{F}^{-1}$ are the Fourier transform and its inverse, respectively. Instead of learning the kernel $\kappa$ in the physical space, FNO directly learns $\mathcal{F}\kappa$ in the frequency domain.

## 4 Coordinate Transform Neural Operators

In FNOs, the kernel integral operator is a convolution operator over the translation group $\mathbb{R}^n$, equipped with its canonical additive group structure. This assumes that the kernel is translation invariant, thereby making the operator itself translation equivariant. Following this idea, FNO can be extended to respect

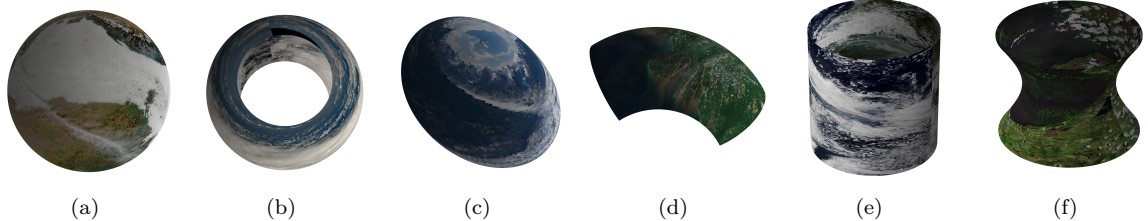

Figure 2: Examples of shapes to which CT-FNO can be applied. (a) Sphere: Polar Coordinates (b) Torus: Polar Coordinates (c) Ellipse: Polar Coordinates (d) Spherical Sector: Polar Coordinates (e) Cylinder: Cylindrical Coordinates (f) Hyperbolic Plane: Hyperbolic Coordinates

other symmetries through canonical coordinate transform. The geometry and boundaries of the domain often dictate certain symmetries that the solution operators might exhibit. For example, if the domain is a square, rotations other than a multiple of 90 degrees do not make sense. Traditional numerical schemes frequently maintain specific symmetry properties of the underlying PDE in different coordinate systems. Symmetries in physics, including PDEs, correlate with the shape and characteristics of the domain. For example, if the domain is circular, it does not make sense to think about (global) translation symmetries anymore. This motivates the pursuit of formulating a coordinate-transform neural operator method that can be applied to different domain shapes and respect the symmetry of the underlying domain shape. In spite of that, we propose the Coordinate Transform Fourier Neural Operators (CT-FNO) framework; CT-FNO is an extension of the FNO architecture in a different coordinate system. A coordinate transformation is first applied to the underlying PDE systems such that the domain shape and symmetries can be natural for FNOs to handle. Then the PDE solution operator is approximated by the neural operator.

### 4.1 Universal Approximation with Coordinate Transform

In this subsection, we provide justification for applying FNO with coordinate transformation. Although it is intuitive that FNO can be applied to different coordinate systems, given that the laws of physics are not dependent on the coordinate system to which they adhere, one may still question the mathematical rigor of such operations. Therefore, we provide the following corollary to justify the use of coordinate transformation in FNOs. A prior work has established the universal approximation theorem for FNOs (Kovachki et al., 2021); importantly, the theorem asserts that, under certain conditions, FNOs maintain continuity as a mapping between Sobolev spaces; thus, FNOs can approximate the behavior of an operator within a specified precision over the considered function space (a given compact subset of a Sobolev Space). A complete treatment of this theoretical aspect is included in Appendix B. In this work, we discuss a modification to FNOs for the purpose of coordinate transformation, which is a parameter mapping; therefore, we present the following corollary.

**Corollary 4.1** (Universal Approximation with Parameter Mapping). *Let $s, s' \geq 0$. Assuming that $\mathcal{G}$, a continuous operator mapping from $H^s(\mathcal{T}^d; \mathbb{R}^{d_a})$ to $H^{s'}(\mathcal{T}^d; \mathbb{R}^{d_u})$, holds true. For any compact subset $K$ contained in $H^s(\mathcal{T}^d; \mathbb{R}^{d_a})$ and any given positive $\epsilon$, there exists a modified Fourier Neural Operator, denoted as $\mathcal{N}'$. This operator $\mathcal{N}'$, mapping from $H^s(\mathcal{T}^d; \mathbb{R}^{d_a})$ to $H^{s'}(\mathcal{T}^d; \mathbb{R}^{d_u})$, incorporates the continuous and compact operators $\mathcal{P}_a$ and $\mathcal{P}_u$ on $H^s(\mathcal{T}^d; \mathbb{R}^{d_a})$ and $H^{s'}(\mathcal{T}^d; \mathbb{R}^{d_u})$, respectively. The structure of $\mathcal{N}'$ is as per equation 10, and it maintains continuity as an operator from $H^s$ to $H^{s'}$, thereby ensuring the following condition is met:*

$$\sup_{a \in K} \|\mathcal{G}(a) - \mathcal{N}'(a)\|_{H^{s'}} \leq \epsilon.$$

This corollary incorporates additional compact operators $\mathcal{P}_a$ and $\mathcal{P}_u$ on the respective Sobolev spaces as parameter mappings (coordinate transform and its inverse), and such compact operators maintain the continuity property of FNOs; thus, the modified FNOs can still approximate the behavior of an operator within a specified precision. Consequently, this corollary enables us to consider CT-FNO by providing theoretical guarantee that CT-FNO can approximate operators in another coordinate system.

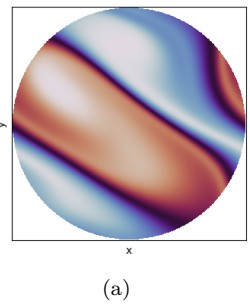 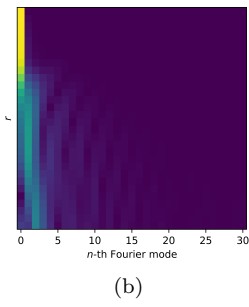 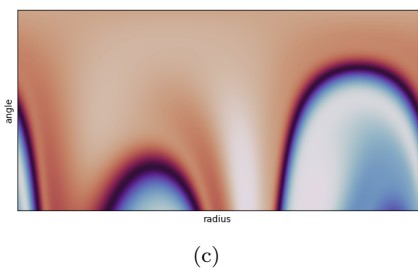

(a)            (b)            (c)

Figure 3: (a): Plot of a function in Cartesian coordinates. (b): The resulting frequency information after applying 1D FFT to the function in polar coordinates for the angular axis. Rows are the coefficients of the Fourier modes of the rings. This is generated from a high-resolution grid in polar coordinates. (c): Plot of the same function in polar coordinates.

## 4.2 Coordinate Transforms and Symmetries

In this subsection, we introduce coordinate transformation and provide an overview of applying coordinate transforms to adapt FNO to various domains while respecting the underlying symmetries. For illustration, we take rotation symmetries and circular domains in 2D as an example. While we focus on 2D circular domains for simplicity, the concept of coordinate transformation readily extends to higher dimensions and other domain shapes. A coordinate transform, in mathematics, is a mapping that relates coordinates in one coordinate system to coordinates in another coordinate system. Symmetries also transform along with coordinate systems. For instance, if the model is isotropic, we would consider convolution over the rotation groups $SO(2)$. To perform convolution over $SO(2)$, an approach outlined in Helwig et al. (2023) involves adapting the workflow of group-equivariant CNNs: rotating the convolution kernels to incorporate a dimension for rotations in the resulting feature maps:

$$(\mathcal{K}v)(R_\phi) = \int_{\mathbb{R}^2} \kappa(R_\phi^{-1}(x,y))v(x,y)dxdy, \tag{3}$$

where $R_\phi = \begin{pmatrix} \cos(\phi) & \sin(\phi) \\ -\sin(\phi) & \cos(\phi) \end{pmatrix} \in SO(2)$.

Letting $x = r\cos(\theta), y = r\sin(\theta)$, the input function $v(x,y)$ and the kernel $\kappa(x,y)$ are transformed to polar coordinates as $v(r,\theta)$ and $\kappa(r,\theta)$, respectively. Although written in a different coordinate system, the underlying physical dynamics remain unchanged. We adapt to the kernel convolution defined in Li et al. (2021) for polar coordinates:

$$(\mathcal{K}v)(r,\theta) = \int \kappa(r-r', \theta-\phi)v(r',\phi)dr'd\theta'. \tag{4}$$

The natural domain shape and symmetry to consider are circular domains with rotation symmetries, denoted by $R_\phi \in SO(2)$. Through the use of canonical coordinates for abelian Lie-groups (Segman et al., 1992), convolution in equation 4 respects the rotation symmetry. It is worth noting that while we have rotational symmetries, it does not make sense to translate the radial axis; in other words, the angular axis is periodic, but the radial axis is not. We apply padding to practically solve this issue. Moreover, the function basis for transformation should be treated differently as these two axes are not equivalent. However, empirically, we observe that the FFT basis is still effective as demonstrated in Section 5.

Therefore, we have extended FNOs to circular domains while respecting the underlying symmetry associated with them. After transforming them into polar coordinates, the domain's shape nicely converts to a rectangle, enabling us to leverage the power of Fast Fourier Transforms (FFTs) to compute the convolution:

$$(\mathcal{K}v)\xi = \mathcal{F}^{-1}(\mathcal{F}\kappa \cdot \mathcal{F}v)\xi, \tag{5}$$

where $\xi$ denotes $(r,\theta)$.

Analogously, FNO can be extended to other domain shapes while respecting the underlying symmetries. As illustrated by examples in Figure 2, CT-FNO can handle various domain shapes. It is important to note that

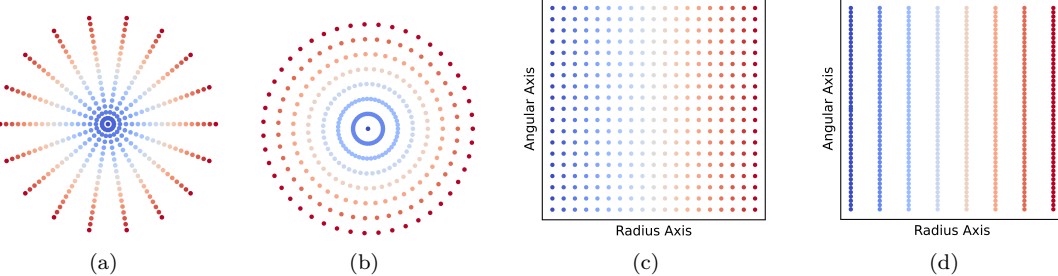

Figure 4: Different Sampling Methods (a): Uniform grid sampling in a ratio of 1 : 1 for the radial and augular axes. (b): Uniform grid sampling in a ratio of 1 : 6. (c)(d): Sampled points in polar coordinates corresponding to (a) and (b), respectively.

this list contains only a small portion of the shapes CT-FNO can operate on. It is not possible to provide a complete list, as there are numerous examples. These shapes, commonly found in numerical PDE literature, have broad applications across various natural sciences and engineering fields (Adler et al., 2013; Leng et al., 2016; Nissen-Meyer et al., 2014). It should be mentioned that there are numerous real-life applications that are naturally modeled in polar coordinates, and achieving rotational equivariance is desirable. Examples of such applications include global weather forecasting, water pipe modeling, electromagnetism, and more. Notably, previous work, such as Bonev et al. (2023), has successfully applied FNO to spherical domains for atmospheric dynamics forecasting. In our study, we generalize the application of FNO to different domain shapes through coordinate transformation. In another line of research, it has been concluded that some symmetries become manifest only in a new coordinate system that must be discovered (Liu & Tegmark, 2022), and group convolution might not be suitable for such symmetries. It is essential to note that CT-FNO has the potential to handle more complex domain geometries or to incorporate hidden symmetries within the underlying PDEs through canonical coordinate transformation.

### 4.3 Sampling Grid for Different Coordinate Systems

In this subsection, we discuss the importance of the sampling grid under different coordinate systems and propose a sampling grid for circular domains as an example to address this issue. Although operator learning is a task involving the mapping of functions, which is grid-independent, oftentimes, we only have access to function values at some finite collocation points. In FNO, to harness the power of FFTs, the sampling grid must be rectangular and equidistant. If the provided grid points are not equidistant, interpolation is applied to obtain uniform grids (equidistant grids) (Liu et al., 2023c). If the PDE is described in a Cartesian coordinate system, and the inference for the solution is uniform in a Cartesian coordinate system, it is critical to choose a sampling grid that represents the function well. Taking polar coordinates as an example, this can also be observed in the frequency domain. If we apply FFT on the angular axis, as shown in Figure 3, the magnitude of high-frequency coefficients increases as $r$ increases. For different applications, different samplings might be preferred, but the message to convey is that sampling grids have to be carefully handled after transformation.

To mediate this issue, we propose sampling an equidistant grid on the radius axis and the angular axis based on a ratio of 1 : 6; this is to respect the geometric nature of a ring in Cartesian coordinate systems, where the ratio of the ring's length to its radius is $2\pi : 1$. For example, if 40 equidistant points are sampled over the radius axis, 240 equidistant points should be sampled over the angular axis. This 1 : 6 ratio is a heuristic made for simplicity; a $1 : 2\pi$ sampling and taking the closest integer for the angular axis can also be used. An illustration of different sampling grids is provided in Figure 4; it is clear that the sampling on the right, which follows the 1 : 6 rule, better represents the function in Cartesian coordinates under a similar number of total sampling points. Although this construction seems simple, it plays a crucial role in reducing the error of interpolation if the points come from a uniform distribution in Cartesian coordinates or an equidistant grid in Cartesian coordinates. We demonstrate this simple yet important aspect in Section 5.3.2.

## 5 Experiments

In this section, we evaluate the proposed CT-FNO based on the following criteria:

- Symmetry Preservation Study: We demonstrate that CT-FNO is effective in preserving symmetries.

- Generalizability Study: We showcase CT-FNO's broader applicability in terms of operating on other domain shapes while respecting the symmetries inherent in the domain shape.

### 5.1 Diffusion of Heat on a Cylinder

In this example, we show that the resulting CT-FNO architecture can work well on domains to which FNO or GFNO cannot be directly applied and outperform baselines that are applicable to such domains. Moreover, we demonstrate CT-FNO's capability to preserve symmetries.

**Simulation Description.** We consider the conduction of heat over time in an isolated medium defined on the surface of a cylinder. There is no heat flux across the top and bottom surfaces of the cylinder. As a result, this system is invariant under translation and equivariant under rotations. We simulate such data with a 3D heat equation on the surface of a cylinder:

$$\frac{\partial U(x,y,z,t)}{\partial t} = \alpha \Delta U(x,y,z,t), \quad (x,y,z,t) \in D \times [0,1]$$
$$\frac{\partial U(x,y,z,t)}{\partial z} = 0, \quad z \in \partial D,$$

where $D := \{(x,y,z) \in \mathbb{R}^3 | x^2 + y^2 = 1, 0 < z < 6\}$, $U$ denotes the temperature, $\alpha = 1$ denotes the thermal diffusivity of the material, $z$ is the axial coordinate, and $t$ denotes time.

**Setup.** We provide details on generating initial conditions in Appendix C. Solutions are obtained by transforming this system into cylindrical coordinates and employing a second-order finite difference scheme for spatial domain and an implicit Euler scheme for time, using a spatial grid of size $128 \times 128$ and a temporal stepsize of 0.01. The resolutions are down-sampled to $64 \times 64$ for training, with point locations stored in both Cartesian and polar coordinates for further usage. A total of 1100 different initial conditions are generated, and the corresponding solutions are obtained. The dataset is divided into 1000 training data samples and 100 testing data samples, referred to as the Normal Testing Set. Additionally, each testing sample in the Normal Testing Set is rotated by a random degree within the discretization tolerance,

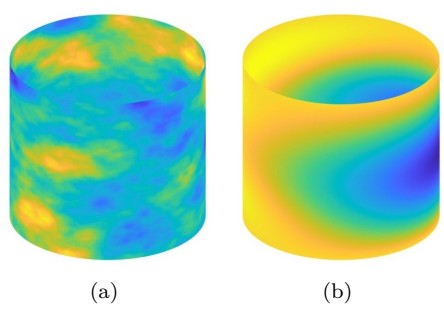

(a)            (b)

Figure 5: (a): Sample Initial Condition (b): Sample Solution at $t = 1$

resulting in the Rotated Testing Set. It is worth noting that rotations will alter the distribution of the data, potentially leading to suboptimal performance due to out-of-distribution issues. Therefore, it is desirable to have an equivariant model to be robust under the Rotated Testing Set. Timing results are recorded on an NVIDIA Tesla V100 32GB GPU. **All results are averaged over 10 different runs, and the mean and standard deviation of relative $\ell_2$ error are reported. This is the setting for all subsequent experiments unless otherwise specified.**

As the origin is fixed for all generated data; in other words, all spatial collocation positions are the same, and there is no translation to be considered. Therefore, even for models that are not translation-invariant, the performance will not be affected negatively. However, CT-FNO does maintain both translation invariance and rotation equivariance as a natural result of coordinate transformation. Since the domain is not rectangular, FNO cannot be directly applied without sacrificing the power FFTs. We compare CT-FNO with baselines applicable to cylindrical surfaces. The results will be compared with GNO (Anandkumar et al., 2019), GINO (Li et al., 2023), and DeepOnet (Lu et al., 2021). For GNO, GINO, and DeepOnets, Cartesian coordinates are used for node connectivity based on radius or as collocation positions. It is worth noting that GNO can also be made equivariant by using relative information, e.g., distances; we included the comparison here. More experimental details can be found in Appendix C, where we also show that this PDE is equivariant to rotations and included more results and details for equivariant GNOs.

Table 1: Results on Diffusion of Heat on a Cylinder. CT-FNO achieves the best performance under the normal testing set and it outperforms all other baselines to a greater extend when symmetries exist. Standard deviations are given in parentheses.

| | Common Information | | | Normal Testing Set | Rotated Testing Set |
|---|---|---|---|---|---|
| | # Par. (M) | Train(%) | Time[a] (s) | Test (%) | Test (%) |
| CT-FNO | 2.18 | 0.147(0.02) | 0.00197 | **0.773**(0.08) | **0.774**(0.08) |
| DeepOnet[b] | 2.40 | 1.508(0.967) | **0.00085** | 1.659(0.992) | 47.783(6.34) |
| GNO | 4.61 | 3.079(1.37) | 0.08062 | 5.885(0.20) | 26.783(4.23) |
| GNO[E] | 4.61 | 2.763(1.29) | 0.08133 | 5.279(0.28) | 5.279(0.28) |
| GINO | 4.63 | 1.386(0.43) | 0.02734 | 1.973(0.51) | 33.374(4.99) |

[a] Evaluation/inference time for a batch of 20 input initial conditions.

[b] Compared to other baselines, DeepOnet is not discritization invariant on the input.

[E] Equivariant Adaptation of GNO; details can be found in Appendix C.2.

**Results and Analysis.** In Table 1, we present the results for the diffusion of heat on a cylinder. CT-FNO outperforms all other baselines in terms of accuracy under the Normal Testing Set. This result is expected, as it is observed in many existing works that FNO outperforms GNO on rectangular domains (Li et al., 2021; Liu et al., 2023a). Compared to DeepOnet, although CT-FNO is slower in inference time, the difference in accuracy is significant. We observe a large standard deviation for DeepOnet; thus, we include results from every single run in Appendix C. We rotate every data sample in the Normal Testing Set to create a new testing set of 100 data samples, denoted as the Rotated Testing Set. Since only CT-FNO is rotation equivariant, it outperforms all other baselines to a greater extent. These results suggest that CT-FNO can extend FNO well to various domain shapes while respecting the underlying symmetries.

## 5.2 Synthetic Operator

In this example, we demonstrate that CT-FNO provides greater generalizability to various domain shapes. Since FNO operates on rectangular domains, we cannot directly compare CT-FNO with FNO or GFNO. However, we consider an inscribed square domain for the other baseline models to demonstrate that the performance of CT-FNO is comparable to all other baselines while providing greater generalizability.

**Simulation Description.** We aim to learn a synthetic operator $G : f \mapsto u$, where $f : \Omega \mapsto \mathbb{R}$ is defined as

$$f(x,y) = \frac{\pi}{K^2} \sum_{i,j=1}^{K} a_{ij} \cdot \left(i^2 + j^2\right)^{-l} \sin(\pi i x) \sin(\pi j y),$$

and $u : \Omega \mapsto R$ is defined as

$$u(x,y) = \frac{1}{\pi K^2} \sum_{i,j}^{K} a_{ij} \cdot \left(i^2 + j^2\right)^{l-1} \sin(\pi i x) \sin(\pi j y),$$

with $l = -0.5, K = 16$, and $a_{ij}$ are i.i.d. uniform on $[-1, 1]$. The dataset is created by setting different sets of random numbers $a_{ij}$.

**Setup.** We define the domain as a circle centered at $(0.5, 0.5)$ with a radius of $\frac{\sqrt{2}}{2}$ for CT-FNO:

$$\Omega := \{(x,y) \in \mathbb{R}^2 : (x - 0.5)^2 + (y - 0.5)^2 < \frac{1}{2}\}.$$

For comparison with other baselines, including FNO, G-FNO, and Radial-FNO, we consider the domain to be a unit box: $\Omega := \{(x,y) \in (0,1)^2\}$, bounded by the circular domain considered in CT-FNO as demonstrated in Figure 6. For the square domain, the grids are $64 \times 64$, and for the circular domain, the grids are $26 \times 156$. We intentionally choose to have a circular domain to show that CT-FNO can operate on circular domains, and its performance is comparable with other baselines on rectangular domains. We bound the square domain with the circular domain for testing purposes. In the shared square, the values are identical given the analytical expressions; if CT-FNO can perform well on the bounding circular domains, it indicates that CT-FNO performs well on the square domain as well. This operator learning task is inspired by a Poisson Equation in Raonic et al. (2023); in fact, for the square domain, $f$ and $u$ are the input function and solution pairs for the Poisson Equation in Appendix D.

The training dataset contains 1000 input-output function pairs, whereas the testing dataset contains 200 input-output function pairs. It is evident that this operator is equivariant to $SO(2)$ rotations; we randomly select 30% and 60% of the testing input-output function pairs and rotate them by an element in the $C_4$ group for square domains and by an arbitrary degree for the circular domain. These rotated pairs are then added to the testing set, referred to as Testing Set A and Testing Set B, respectively.

**Results and Analysis.** In Table 2, we present results for the Synthetic Operator. Considering that the square domain for other baselines is bounded by the circular domain for CT-FNO, we may conclude that CT-FNO achieves roughly the same level of accuracy compared to all other baselines. It can be seen that CT-FNO is able to operate on circular domains while respecting the underlying rotational symmetries without a noticeable increase in computational costs. Although we do not observe superior performance of CT-FNO over other baselines, our contribution focuses on the generalizability of CT-FNO to extend FNO to other domain shapes while maintaining the underlying symmetries. Moreover, with such generalization, CT-FNO is robust under arbitrary rotations as the discretization tolerates, whereas other baselines are constrained to $C_4$ rotations due to the nature of

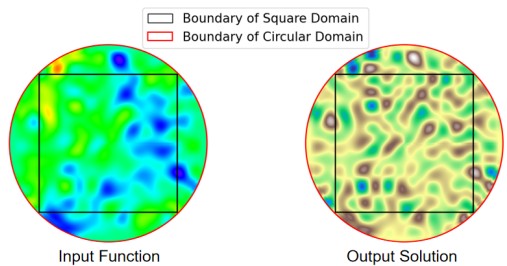

Figure 6: Sample input function and output solution. Input function and output solution values are the same in the shared square.

square domains. The experimental findings in this example corroborate with the conclusion from Section 4.1 that CT-FNO, similar to FNO, can approximate operators in another coordinate system with theoretical guarantees.

Table 2: Results on 2D Synthetic Operator. CT-FNO achieves rotational equivariance without significantly increasing computational complexity; the training time per epoch is close to that of FNO and much less than that of G-FNO. Moreover, it can be seen that CT-FNO can operate on circular domains while maintaining similar performance as all other baselines. Standard deviations are given in parentheses.

|  | Common Information | | | Testing Set A | Testing Set B |
| --- | --- | --- | --- | --- | --- |
|  | # Par. (M) | Train(%) | TPE[a] (s) | Test (%) | Test (%) |
| FNO ($C_4$) | 2.37 | 0.412(0.03) | **0.416** | 22.537(1.53) | 31.942(1.53) |
| G-FNO ($C_4$) | 2.24 | **0.378(0.04)** | 0.709 | **0.400**(0.04) | **0.402**(0.04) |
| Radial-FNO ($C_4$) | 2.63 | 0.386(0.03) | 0.540 | 0.431(0.04) | 0.434(0.04) |
| CT-FNO (Arbitrary) | 2.31 | 0.399(0.06) | 0.459 | **0.403**(0.07) | **0.402**(0.06) |

[a] TPE: Average **T**ime **P**er **E**poch since G-FNO will increase the channel width and increase FFT computations.

### 5.3  2D Darcy Flow Equation

#### 5.3.1  Symmetry Study

In this example, we specifically demonstrate CT-FNO's capability to preserve rotation symmetries, even for rectangular domains.

**Simulation Description.** The Darcy flow equation is a fundamental equation in fluid dynamics that describes the flow of fluids through porous media. It is crucial in various fields like hydrogeology, petroleum engineering, soil mechanics, and environmental science. We consider the steady-state of the 2D Darcy Flow equation from Li et al. (2021) given by:

$$-\nabla \cdot (a(x)\nabla u(x)) = f(x) \quad x \in (0,1)^2$$
$$u(x) = 0 \qquad x \in \partial(0,1)^2$$

where $a \in L^\infty\left((0,1)^2; \mathbb{R}_+\right)$ is the diffusion coefficient and $f \in L^2\left((0,1)^2; \mathbb{R}\right)$ is the forcing function that is kept fixed $f(x) = 1$. We are interested in learning the operator mapping the diffusion coefficient $a(x)$ to the solution $u(x)$. It can be shown that this operator is equivariant to the $C_4$ rotation group (rotational symmetry group of order 4) as shown in Appendix E.

**Setup.** The input diffusion coefficient field $a(x, y)$ is generated by the Gaussian random field with a piecewise function, namely $a(x, y) = \psi(\mu)$, where $\mu$ is a distribution defined by $\mu = \mathcal{N}\left(0, (-\Delta + 9I)^{-2}\right)$. The mapping $\psi : \mathbb{R} \rightarrow \mathbb{R}$ takes the value 12 on the positive and 3 on the negative, and the push-forward is defined pointwise. Solutions are obtained using a second-order finite difference scheme on a $241 \times 241$ grid and then downsampled to $49 \times 49$. For CT-FNO, we use bilinear interpolation to convert Cartesian grids into $40 \times 240$ equidistant grids in polar coordinates.

For this 2D Darcy Flow, we apply CT-FNO to achieve rotation equivariance. As the domain is a unit box, we test rotation equivariance under the $C_4$ group. The results will be compared with FNO (Li et al., 2021), G-FNO (Helwig et al., 2023), and radial-FNO (Shen et al., 2022; Helwig et al., 2023), which is a frequency domain radial kernel that is invariant to rotations. The data comes in an equidistant grid in Cartesian coordinates. To apply CT-FNO, we convert it to polar coordinates using bi-linear interpolation, and the solution will be interpolated back to Cartesian coordinates through bi-linear interpolation. Since the resulting domain is not rectangular in polar coordinates, reflection padding is applied. The reported errors are based on Cartesian coordinates. The training dataset contains 1000 input-output function pairs, whereas the testing dataset contains 200 input-output function pairs, denoted as the Normal Testing Set. We rotate every testing sample in the Normal Testing Set by an element in the $C_4$ group and denote this testing dataset as the Rotated Testing Set to test rotation equivariance.

Table 3: Results on the 2D Darcy Flow equation. CT-FNO achieves the best performance without significantly increasing computational complexity; the training time per epoch is close to that of FNO. We rotate the testing input-output function pairs and rotate by an element in the $C_4$ group; these testing data-set is called the Rotated Testing Set. Standard deviations are given in parentheses.

| | Common Information | | | Normal Testing Set | Rotated Testing Set |
|---|---|---|---|---|---|
| | # Par. (M) | Train(%) | TPE[a] (s) | Test (%) | Test (%) |
| FNO | 2.37 | 0.335(0.03) | **0.413** | 0.701(0.04) | 1.765(0.10) |
| G-FNO | 2.24 | 0.315(0.03) | 0.710 | 0.693(0.05) | 0.693(0.05) |
| Radial-FNO | 2.63 | 0.331(0.04) | 0.539 | 0.713(0.08) | 0.713(0.08) |
| CT-FNO | 2.31 | 0.276(0.06) | 0.462 | **0.664**(0.10) | **0.669**(0.10) |

[a] TPE: We record the average **t**ime **p**er **e**poch since G-FNO increases the channel width and increase FFT computations.

**Results and Analysis.** In Table 3, we present the results for the Darcy Flow equation. CT-FNO performs slightly better compared to all other baselines under normal data conditions. However, it does not provide enough evidence that CT-FNO performs better than other baselines under normal testing data, and this is expected as the objective of CT-FNO is to target symmetries and generalizability to other shapes for which FNO cannot be directly applied. For $C_4$ testing data, the performance of FNO degrades moderately, although not as obviously as in previous examples as the coefficient functions are sampled from Gaussian random fields and then mapped to piecewise constants, and thus, the distribution of the rotated testing set does not deviate much from the normal testing set.

Table 4: Results for CT-FNO on 2D Darcy Flow under different grid samplings. The more it deviates from the proposed grid sampling, the larger the error becomes. CT-FNO is trained on grid values in polar coordinates obtained by interpolation.

| Sampling Grid Size | Test (%) | Train (%) |
|---|---|---|
| $20 \times 120^{*}$ | 0.687 | 0.276 |
| $30 \times 80$ | 0.738 | 0.301 |
| $40 \times 60$ | 0.847 | 0.331 |
| $49 \times 49$ | 0.966 | 0.354 |
| $15 \times 160$ | 0.754 | 0.359 |
| $12 \times 200$ | 0.873 | 0.432 |
| $10 \times 240$ | 1.057 | 0.501 |
| $8 \times 300$ | 1.361 | 0.616 |

[*] Proposed Sampling Grid

CT-FNO, G-FNO, and radial-FNO are all equivariant to rotations; thus, their performances are robust against rotations in the testing set. The testing errors do not change for G-FNO and Radial-FNO; the testing error varies slightly for CT-FNO due to interpolation applied. Equivariant architectures can extract local symmetries (Cohen & Welling, 2016; 2017) and thus improve results even on data without global symmetries. However, in FNO-based architectures, convolution kernels are global; it is unclear whether group global convolution can capture local symmetries. The group-equivariant architecture can still enhance

network performance as it provides broader expressive power to the neural network in the frequency domain; however, this increase also raises the computational complexity of FFT due to the larger number of channels.

### 5.3.2 Ablation Study on Sampling Grid

We compare the results under different sampling grids as discussed in Section 4.3. Bilinear interpolation is applied for all cases. In Table 4, we demonstrate the impact of using different sampling grids. It can be observed that the proposed $1:6$ rule exhibits superior performance compared to other sampling grids. As the ratio deviates further from $1:6$, the performance decreases monotonically. Therefore, we conclude that the proposed sampling grid is highly effective and important. Analogously, for other domain shapes and coordinate transformations, one can identify a sampling grid similar to this to respect the geometric nature of the Cartesian coordinate system. For this example, the results are averaged over 5 runs.

## 6 Conclusion, Limitations, and Future Work

In this work, we propose designing an FNO architecture based on coordinate transformation. Specifically, by considering the underlying shape and symmetries of the domains it embeds, we may apply a coordinate transformation to seamlessly convert the domain into a rectangular domain in which FNO can be applied. Through coordinate transform, we convert rotational symmetries into translation symmetries, which are naturally inherent in FNO architectures. We conduct experiments to evaluate our proposed CT-FNO. Results show that operator learning with coordinate transformation can achieve similar performance to FNO while being able to generalize to various domain shapes while respecting the symmetries.

**Limitations.** Although this work generalizes FNO to various domains, it is still not broad enough; it would be interesting to see if coordinate transformation can be applied with other methods to work on general domains while respecting the symmetries.

**Future Work.** In Bonev et al. (2023), the authors have explored the use of a different coordinate system for climate modeling. As our work is a generalization of coordinate transformation for neural operators, it will be interesting to see if CT-FNO can be applied to more practical real physical systems. Moreover, some symmetries become manifest only in a new coordinate system that must be discovered (Liu & Tegmark, 2022); it is worth exploring whether such symmetries can be captured by CT-FNO.

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

# A  Group Equivariance Preliminaries

**Definition A.1.** A **group** is an algebraic structure that consists of a set $G$ together with a binary operator, $\cdot$, the group product, that satisfies the following axioms:

- Closure: For all $h, g \in G$ we have $h \cdot g \in G$.

- Identity: There exists an identity element $e$.

- Inverse: for each $g \in G$ there exists an inverse element $g^{-1} \in G$ such that $g^{-1} \cdot g = g \cdot g^{-1} = e$.

- Associativity: For each $g, h, i \in G$ we have $(g \cdot h) \cdot i = g \cdot (h \cdot i)$.

**Definition A.2.** A subset $H \subseteq G$ of a group $G$ forms a **subgroup** if it is closed under composition and taking inverses:

- Composition: for all $g, h \in H$ one has $gh \in H$

- Inversion: for all $g \in H$ one has $g^{-1} \in H$

Subgroups are also groups; in other words, they satisfy the group axioms.

**Definition A.3.** Let $(G, \cdot_G)$ and $(H, \cdot_H)$ be two groups. Their (outer) **direct product** $(G, \cdot_G) \times (H, \cdot_H)$ is defined on the Cartesian product $G \times H$ of the underlying sets, equipped with the group product $\cdot$ defined as

$$G \times H \to G \times H, \quad (g_1, h_1) \cdot (g_2, h_2) = (g_1 \cdot_G g_2, h_1 \cdot_H h_2)$$

with the condition that the elements of the factors $G$ and $H$ are independent from each other. If the group $H$ acts on $G$, the notion of direct product groups is generalized to semi-direct product groups, denoted as $G \rtimes H$.

In this work, we have mentioned several special groups. Herein, we provide the definitions for the general audience:

- The Orthogonal group $O(n)$ is the group of orthogonal matrices in $n$ dimensional Euclidean space. These matrices preserve the dot product, representing rotations and reflections.

- The Special Orthogonal group $SO(n)$ is a subgroup of $O(n)$ and consists of proper rotations in $n$-dimensional Euclidean space without reflections.

- The Euclidean group $E(n) = \mathbb{R}^n \rtimes O(n)$ is the group of isometries of Euclidean space $\mathbb{R}^n$, which includes translations and rotations.

- The Special Euclidean group $SE(n) = \mathbb{R}^n \rtimes SO(n)$ is a subgroup of $E(n)$ and consists of rigid transformations, which include translations and rotations, but the rotations are limited to proper rotations (no reflections).

- The cyclic group $C_4$, also known as the cyclic group of order 4 , is defined as the group consists of four elements, denoted by $C_4 = \{R_0, R_{90}, R_{180}, R_{270}\}$, representing rotations of $0°, 90°, 180°$, and $270°$, repetitively.

- The group $p4$ consists of all compositions of translations and rotations in the $C_4$ group.

**Definition A.4.** Let $G$ be a group and $X$ be a set. A (left) group action is a mapping from $G \times X$ to $X$ that satisfies the following properties:

- Identity Element: $e \cdot x = x$ for all $x \in X$, where $e$ is the identity element of $G$.

- Compatibility: $(gh) \cdot x = g \cdot (h \cdot x)$ for all $g, h \in G$ and $x \in X$.

# B    Theoretical Analysis of Parameter Transformation

The concept of FNOs, as introduced in the seminal work (Li et al., 2021), holds significant relevance in the context of the periodic domain $\mathcal{T}^d$. In this setting, FNOs are conceptualized as a mapping $\mathcal{N} : \mathcal{A}(D; \mathbb{R}^{d_a}) \rightarrow \mathcal{U}(D; \mathbb{R}^{d_u})$. The formulation of this mapping is as follows:

$$\mathcal{N}(a) := \mathcal{Q} \circ \mathcal{L}_L \circ \mathcal{L}_{L-1} \circ \cdots \circ \mathcal{L}_1 \circ \mathcal{R}(a), \tag{6}$$

where $\mathcal{R}$ and $\mathcal{Q}$ represent the lifting and projection operators, respectively, as elaborated in equation 8 and equation 9. Further, the non-linear layers $\mathcal{L}_\ell$ within this structure are represented by

$$\mathcal{L}_\ell(v)(x) = \sigma\left( W_\ell v(x) + b_\ell(x) + \mathcal{F}^{-1}\left( P_\ell(k) \cdot \mathcal{F}(v)(k) \right)(x) \right). \tag{7}$$

Here, $W_\ell \in \mathbb{R}^{d_v \times d_v}$ and $b_\ell(x)$ constitute a pointwise affine mapping (reflecting weights and biases), while $P_\ell : \mathbb{Z}^d \rightarrow \mathbb{C}^{d_v \times d_v}$ specifies the coefficients for a non-local, linear mapping via the Fourier transform.

Moreover, the lifting operator $\mathcal{R} : \mathcal{A}(D; \mathbb{R}^{d_a}) \rightarrow \mathcal{U}(D; \mathbb{R}^{d_v})$, where $d_v \geq d_u$, functions locally and is defined as

$$\mathcal{R}(a)(x) = Ra(x), \quad R \in \mathbb{R}^{d_v \times d_a}, \tag{8}$$

and the projection operator $\mathcal{Q} : \mathcal{U}(D; \mathbb{R}^{d_v}) \rightarrow \mathcal{U}(D; \mathbb{R}^{d_u})$ operates locally, given by

$$\mathcal{Q}(v)(x) = Qv(x), \quad Q \in \mathbb{R}^{d_u \times d_v}. \tag{9}$$

An adaptation to the FNO architecture is proposed by integrating two additional parameter mapping modules. This leads to a revised mapping $\mathcal{N}' : \mathcal{A}(D; \mathbb{R}^{d_a}) \rightarrow \mathcal{U}(D; \mathbb{R}^{d_u})$, expressed as

$$\mathcal{N}'(a) := \mathcal{P}_u \circ \mathcal{Q} \circ \mathcal{L}_L \circ \mathcal{L}_{L-1} \circ \cdots \circ \mathcal{L}_1 \circ \mathcal{R} \circ \mathcal{P}_a(a), \tag{10}$$

where $\mathcal{P}_a : \mathcal{A}(D; \mathbb{R}^{d_a}) \rightarrow \mathcal{A}(D; \mathbb{R}^{d_a})$ and $\mathcal{P}_u : \mathcal{U}(D; \mathbb{R}^{d_u}) \rightarrow \mathcal{U}(D; \mathbb{R}^{d_u})$ function as parameter mappings.

In accordance with the foundational principles outlined in the universal approximation theorem for FNOs (Kovachki et al., 2021), we extend these concepts through the following theorem:

**Theorem B.1** (Universal Approximation for FNOs). *Let us consider two non-negative integers, $s$ and $s'$, where both $s, s' \geq 0$. Let $\mathcal{G}$ be a continuous operator, defined from $H^s(\mathcal{T}^d; \mathbb{R}^{d_a})$ to $H^{s'}(\mathcal{T}^d; \mathbb{R}^{d_u})$. Given any compact subset $K$ within $H^s(\mathcal{T}^d; \mathbb{R}^{d_a})$ and for any arbitrary positive value of $\epsilon$, there exists a Fourier Neural Operator, denoted as $\mathcal{N}$, mapping from $H^s(\mathcal{T}^d; \mathbb{R}^{d_a})$ to $H^{s'}(\mathcal{T}^d; \mathbb{R}^{d_u})$. This operator $\mathcal{N}$, structured according to the form delineated in equation 6, maintains continuity as a mapping from $H^s$ to $H^{s'}$. Consequently, this results in the following inequality being satisfied:*

$$\sup_{a \in K} \|\mathcal{G}(a) - \mathcal{N}(a)\|_{H^{s'}} \leq \epsilon.$$

Building upon this theorem, we derive a corollary, that follows immediately from Theorem B.1, to address the modifications involving parameter mapping within the FNO framework:

**Corollary B.2** (Universal Approximation with Parameter Mapping). *Let $s, s' \geq 0$. Assuming that $\mathcal{G}$, a continuous operator mapping from $H^s(\mathcal{T}^d; \mathbb{R}^{d_a})$ to $H^{s'}(\mathcal{T}^d; \mathbb{R}^{d_u})$, holds true. For any compact subset $K$ contained in $H^s(\mathcal{T}^d; \mathbb{R}^{d_a})$ and any given positive $\epsilon$, there exists a modified Fourier Neural Operator, denoted as $\mathcal{N}'$. This operator $\mathcal{N}'$, mapping from $H^s(\mathcal{T}^d; \mathbb{R}^{d_a})$ to $H^{s'}(\mathcal{T}^d; \mathbb{R}^{d_u})$, incorporates the continuous and compact operators $\mathcal{P}_a$ and $\mathcal{P}_u$ on $H^s(\mathcal{T}^d; \mathbb{R}^{d_a})$ and $H^{s'}(\mathcal{T}^d; \mathbb{R}^{d_u})$, respectively. The structure of $\mathcal{N}'$ is as per equation 10, and it maintains continuity as an operator from $H^s$ to $H^{s'}$, thereby ensuring the following condition is met:*

$$\sup_{a \in K} \|\mathcal{G}(a) - \mathcal{N}'(a)\|_{H^{s'}} \leq \epsilon.$$

# C Diffusion of Heat

## C.1 Initial Conditions

We generate the initial conditions

$$U(\xi,0) = U_0(\xi) = U_0(x,y,z) = \sum_{n=-N}^{N} \sum_{m=-M}^{M} [A_{nm} \cos(n \arccos x) + B_{nm} \sin(n \arcsin y)] \cos\left(\frac{m\pi z}{L_z}\right) \cdot S_{nm}$$
$$+ \frac{\pi}{5} \exp\left(\frac{-10(\arccos x - \pi)^2}{2\pi}\right) + 1,$$

where $A_{nm}, B_{nm} \sim \mathcal{N}(0, 0.5)$, $S_{nm} = \tau^{(\gamma-1)} \cdot \left(\pi^2 \lambda_{nm} + \tau^2\right)^{-\frac{\gamma}{2}}$ with $\tau = 1$ and $\gamma = 2$, and $\lambda_{nm} = \left(\frac{n^2}{L_\theta^2} + \frac{m^2}{L_z^2}\right)$ with $L_\theta = 2\pi$ and $L_z = 6$. The initial temperature generated satisfies periodic conditions for continuity around the cylinder's circumference. These initial temperatures distributions are made to demonstrate the effectiveness and importance of the equivariant nature of CT-FNO. The second term introduces a higher initial temperature at fixed locations on the cylinder; consequently, rotations will alter the initial heat distribution. In various real-world applications, rotations will have a significant impact as well.

## C.2 Details and Additional Results on Equivariant Adaptation of GNO

For the implementation of GNO, we directly adopt the settings from Anandkumar et al. (2019). In this approach, the node features, denoted as $f_i$ for node $i$, comprise the function value and the positional vector in Euclidean space ($x, y, z$ coordinates) associated with the node. Similarly, the edge features, denoted as $e_{ij}$ for the edge between nodes $i$ and $j$, encompass all the function values and positional vectors of nodes $i$ and $j$.

How to design more sophisticated equivariant GNNs is a field of study (Satorras et al., 2021; Liu et al., 2023b; 2022). Moreover, lately, there have been studies on more sophisticated designs of GNOs or their variants, such as Liu et al. (2023b); Xu et al. (2024), for certain tasks. A straightforward equivariant adaptation of GNO, which we use to compare with CT-FNO, is to use a reference to embed geometric information, in a way that respects the underlying symmetries, in the node and edge features. For node $i$, we denote its coordinates by $x_i$, and we take the two points with the maximum and minimum function values, respectively, among its neighbors defined by the cut-off radius as reference points. We denote these two points as $x_{\min}$ and $x_{\max}$. Now, the node feature of the $i$th node, $f_i$, comprises the function value and the distances between node $i$ and the two reference points:

$$d_i^{\min} = ||\vec{x}_i - \vec{x}_{\min}||, \text{ and } d_i^{\max} = ||\vec{x}_i - \vec{x}_{\max}||.$$

For edge features, $e_{ij}$, between nodes $i$ and $j$, we include the four distances $[d_i^{\min}, d_j^{\min}, d_i^{\max}, d_j^{\max}]$ and the pairwise distance, $d_{ij} = ||\vec{x}_i - \vec{x}_j||$, between nodes $i$ and $j$. It is clear that such construction of node and edge features is equivariant to rotations. Therefore, GNO is made equivariant.

## C.3 Rotation Equivariance

**Lemma C.1.** *Let $D = \{(x,y,z) \in \mathbb{R}^3 | x^2 + y^2 = 1, 0 < z < 6\} \subseteq \mathbb{R}^3$ be the cylindrical domain considered with the center of the base being the origin and $T \subset \mathbb{R}_{\geq 0}$ be the space of time. Suppose that the functions $U : D \times T \to \mathbb{R}$ solve the partial differential equation*

$$\frac{\partial U(x,y,z,t)}{\partial t} = \alpha \Delta U(x,y,z,t), \quad (x,y,z,t) \in D \times T$$
$$\frac{\partial U(x,y,z,t)}{\partial z} = 0, \quad z \in \partial D, \tag{11}$$

Table 5: Test and train errors from all runs for DeepOnet vary greatly. However, even in the case of the best result, it is still slightly less accurate than FNO.

| | Test Error | Train Error |
|---|---|---|
| 1 | 0.00996 | 0.00799 |
| 2 | 0.00998 | 0.00817 |
| 3 | 0.01003 | 0.00821 |
| 4 | 0.01033 | 0.00866 |
| 5 | 0.01040 | 0.01008 |
| 6 | 0.01097 | 0.01021 |
| 7 | 0.01138 | 0.01036 |
| 8 | 0.03011 | 0.02798 |
| 9 | 0.03129 | 0.02954 |
| 10 | 0.03146 | 0.02957 |

*Take $R_\theta$ to be the representation of a rotation around z-axis acting on D, which can be described by unitary matrices:*

$$R_\theta = \begin{bmatrix} \cos\theta & -\sin\theta & 0 \\ \sin\theta & \cos\theta & 0 \\ 0 & 0 & 1 \end{bmatrix}$$

*Then the function $U_R : R_\theta D \times T \to \mathbb{R}$ also satisfy equation 11 for all $\theta \in [0, 2\pi]$, where $U_R(x, y, z, t) := U(R_\theta(x, y, z), t)$*

*Proof.* Equation 11 can be written as:

$$\frac{\partial U(\beta, z, t)}{\partial t} = \alpha \left( \frac{1}{r^2} \frac{\partial^2 U(\beta, z, t)}{\partial \beta^2} + \frac{\partial^2 U(\beta, z, t)}{\partial z^2} \right)$$
$$\frac{\partial U(\beta, z, t)}{\partial z} = 0 \tag{12}$$

where $\beta = \arccos x = \arcsin y \in [0, 2\pi]$. The representation of rotation around z-axis will just result in a periodic translation in $\theta$. Thus, it suffices to show that if $U(\beta, z, t)$ satisfies equation 12, then $U(z, \beta - \theta, t)$ also satisfies equation 12, which is trivial.

$\square$

### C.4 DeepOnet Results

The results for DeepOnet from every single run are recorded in Table 5.

## D Poisson Equation

This prototypical linear elliptic PDE from Raonic et al. (2023) is given by,

$$-\Delta u = f, \text{ in } D, \quad u|_{\partial D} = 0.$$

The solution operator $\mathcal{G} : f \mapsto u$, maps the source term $f$ to the solution $u$. With source term,

$$f(x, y) = \frac{\pi}{K^2} \sum_{i,j=1}^{K} a_{ij} \cdot (i^2 + j^2)^{-r} \sin(\pi i x) \sin(\pi j y), \quad \forall (x, y) \in D,$$

with $r = -0.5$, $D := \{(x, y) \in (0, 1)^2\}$, the corresponding exact solution can be analytically computed:

$$u(x, y) = \frac{1}{\pi K^2} \sum_{i,j}^{K} a_{ij} \cdot \left(i^2 + j^2\right)^{l-1} \sin(\pi i x) \sin(\pi j y),$$

we fix $K = 16$ and choose $a_{ij}$ to be i.i.d. uniformly distributed from $[-1, 1]$.

## E    Rotation Equivariance of the Darcy Flow Equation

**Lemma E.1.** *Let $D \subseteq \mathbb{R}^2$ be a domain. Suppose that the functions $u : D \to \mathbb{R}, a : D \to \mathbb{R}_+$, and $f : D \to \mathbb{R}$ solve the partial differential equation*

$$-\nabla \cdot (a(x)\nabla u(x)) = f(x) \tag{13}$$

*Take $R$ to be an orthogonal matrix describing a rotation. Then the functions $u_R : R^{-1}D \to \mathbb{R}, a_R : R^{-1}D \to \mathbb{R}_+$, and $f_R : R^{-1}D \to \mathbb{R}$ also satisfy equation 13, where:*

$$\begin{aligned} u_R(x) &:= u(Rx) \\ a_R(x) &:= a(Rx) \\ f_R(x) &:= f(Rx) \end{aligned} \tag{14}$$

Note that Lemma E.1 is most meaningful when the domain $D$ is invariant under the rotation $R$, e.g., the domain is a circle or the domain is $\mathbb{R}^2$. If the domain is a unit box, $D = (0, 1)^2$, it is most meaningful to consider rotation by $0°, 90°, 180°$ or $270°$.

*Proof.* Since $f_R(x) = f(x) = 1$ is invariant under rotation, to show that $u_R, a_R$, and $f_R$ satisfy equation 13, it is sufficient to show that $\nabla \cdot (a_R \nabla u_R)(x) = (\nabla \cdot a\nabla u)(Rx)$. We have

$$\begin{aligned} \nabla \cdot (a_R(x)\nabla u_R(x)) &= \sum_{i=1}^{2} \frac{\partial}{\partial x_i}\left(a_R(x)\frac{\partial}{\partial x_i}u_R(x)\right) \\ &= \sum_{i=1}^{2} \frac{\partial}{\partial x_i}\left(a(Rx)\frac{\partial}{\partial x_i}u(Rx)\right) \\ &= \sum_{i=1}^{2}\sum_{j=1}^{2}\sum_{k=1}^{2}\left(\frac{\partial a}{\partial x_k}(Rx)\frac{\partial u}{\partial x_j}R_{ki}R_{ji} + a(Rx)\frac{\partial u}{\partial x_k}\frac{\partial u}{\partial x_j}R_{ki}R_{ji}\right) \\ &= \frac{\partial a}{\partial x_1}(Rx)\frac{\partial u}{\partial x_1}(Rx) + a(Rx)\frac{\partial^2 u}{\partial x_1^2}(Rx) + \frac{\partial a}{\partial x_2}(Rx)\frac{\partial u}{\partial x_2}(Rx) + a(Rx)\frac{\partial^2 u}{\partial x_2^2}(Rx) \\ &= (\nabla \cdot a\nabla u)(Rx). \end{aligned} \tag{15}$$

$\square$

