# OpenReview forum: "Coordinate Transform Fourier Neural Operators for Symmetries in Physical Modelings"
_TMLR — Accepted by TMLR_

### Review · Reviewer_mw3p · 2024-05-02

**Summary Of Contributions:**

The authors present an approach to apply Fourier neural operators (FNO) to non-euclidian data domains, by adding fixed coordinate transforms mapping the data domain to a rectangular input domain of the FNO, and analogously from the FNO output to the target domain. The proposed method is in particular useful to map the translation equivariance of the FNO model to another equivariance arising from the symmetry of the domain, such as a rotation symmetry. Tests with polar and cylindrical coordinates are shown.

I am not closely familiar with the large existing literature on physics-informed ML, and will therefore not comment on the novelty and contribution of the approach.

**Audience:**

Yes

**Broader Impact Concerns:**

/

**Claims And Evidence:**

Yes

**Requested Changes:**

### Main requested changes or clarifications


1. **Design of the experiments in sect.5.1 (Cylinder)**
 - The authors state for the cylinder: "Since the domain is not rectangular,
FNO cannot be directly applied without sacrificing the power FFTs."
    However, the direct method here seems to be to cut vertically and flatten the cylinder, giving a rectangular grid. And since a FFT assumes periodic boundary conditions, a FNO would be rotationally equivariant both in the angular direction (correctly) and in the vertical direction (incorrectly). Is this not also what is done by the CT-FNO in this case? Possibly I am misunderstanding the setup here.
 - It is stated that the GNO can also be used such that it is equivariant - then this should be done and added to the comparison.
 - It was unclear to me why the performance on the rotated test set should be worse than on the normal test set for the comparison methods. Are the initial conditions of the training set not generated uniformly across angles?
 - How are the vertical boundaries of the grid in CT-FNO treated compared to the angular (periodic) boundary?



### Optional and minor changes

2. **Vague abstract**
  - The abstract remained relatively vague, explaining the concept of symmetries in physics, but very little about the concrete approach and its window of applicability. My recommendation is to rewrite the abstract to be more precise and descriptive.
  - In the last sentence, the scheme "generalizes well across different domain shapes" seems to suggest that a single model could be trained on one domain shape, and then generalize at test time to a new domain shape. (The formulation also reappears in the main text). However, such an experiment, while an interesting possibility, is not part of the presented results.

3. **Missing general definition of CT-FNO**
 - A general statement of how the coordinate transform is added to the FNO in section 4 would be useful.

3. **Sampling grid and Fig.4**
  - Fig. 4 was visually confusing: panel a) corresponds to d), and panel b) to c) ?
  - Why are the circular domains mapped to square domains? If the aim is to proportionally represent radial and angular axes, measured by the outside circumference of the circle, why not represent this as a rectangle with aspect ratio $1 : 2 \pi$ ?
  - This figure highlights how uniform sampling on the coordinate grid can lead to non-uniform resolution in the physical embedding space (close compared to far from the center of the circle). Could the authors discuss when this is a useful or problematic feature? For example, in a rotating physical system the characteristic angular size of feature fluctuations may vary along the radial coordinate.

4. **Omitted proof and unclear significance of Corollary 4.1**
 - Both the main text and Appendix D state Corollary 4.1 (respectively D.2) at length, without explict proof since it would follow immediately from the theorem of Kovachki et al. '21. Since the authors appear to nonetheless assign importance to the extension and do not simply cite the result of Kovachki et al. '21, it might be good to formalize the logic of adding the coordinate transforms in a short proof.


5. **Discussion of discovering symmetries**
 - The conclusion raises the question whether CT-FNO could capture and discover symmetries which are not known a priori. However, the coordinate transformation must be chosen by hand in the current approach. Is the idea here to make the coordinate transform learnable?


6. **Typos**
  - Sect. 5.1: "More details of experimental details"
  - App. E: "Poisson Equation" appears doubled

**Strengths And Weaknesses:**

### Strengths

- Generally clear manuscript
- Didactic introduction of group convolution and Fourier neural operators

### Weaknesses

- Emphasis that CT-FNO is applicable beyond rotations and in higher dimensional domains, but no experiments demonstrating this

---

> ### Author Response · Authors · 2024-05-29
> **Reply to Reviewer mw3p by Authors (1st Part)**
>
> We deeply appreciate your time and effort in reviewing our work, and we value the insights and feedback you've provided. We have made necessary changes based on your requests, including the optional ones. We have revised the paper and added new experiments as you requested. We address your comments below.
> ### Regarding Weakness of Our Work
> > Emphasis that CT-FNO is applicable beyond rotations and in higher dimensional domains, but no experiments demonstrating this
> - Thank you very much for your comments. We acknowledge that many of the domains we present in our work are effectively circular and possess rotational symmetries. In the paper, we transparently state this limitation and explicitly mention that **extending the applicability of the FNO to broader contexts is reserved for future work (e.g. Limitations under Sec. 6)**. We hope this addresses your concerns.
> - Regarding higher-dimensional applications, we understand this to refer to the dimensionality of the underlying PDE, but correct us if we understood your concern incorrectly.  In the first paragraph of Section 4.2, we clarify that “While we focus on 2D circular domains for simplicity, the concept of coordinate transformation readily extends to higher dimensions and other domain shapes.” **This focus is chosen to simplify notation and explanation, the concept extends readily to higher dimensions as convolution itself can be defined in higher dimensions, so does coordinate transformations. Notably, we do have an example involving 3D space in our experiments (Sec. 5.1 Cylinder).**
> - We did not conduct experiments in even higher dimensions (4D and higher) due to the computational expense associated with data generation and training; also, it is a little out of the scope of this work. **However, the extension of our methods to higher dimensions is straightforward, particularly for shapes within the scope of this work. For instance, rotational symmetries in hyperspheres (higher-dimensional spheres) can be transformed into translational symmetries in cubes or hypercubes**. We are confident that exploring these higher-dimensional applications is feasible for the discussions within the scope of this work.
> - If you are referring to the challenges associated with high-dimensional PDEs beyond our focus on coordinate transformations for domain shapes and symmetries, it is important to note that FNO itself suffers from the so-called “curse of dimensionality.” As the dimensions increase, the computational complexity grows exponentially. This issue is also observed in most numerical methods for PDEs. **That being said, our primary focus is not on mitigating the curse of dimensionality in FNO, but rather on explaining why we chose 2D domains to illustrate our method.** Our approach is indeed applicable to higher dimensions (e.g., hyperspheres) provided there are sufficient computational resources.
> - We greatly value your feedback and suggestions and will strive to address your concerns. We hope this clarifies our position and addresses your concerns regarding the limitations or weaknesses of our work. If you have any further concerns regarding this, please let us know, and we will do our best to address them.

---

> ### Author Response · Authors · 2024-05-29
> **Reply to Reviewer mw3p by Authors (2nd Part)**
>
> ### Regarding Main Requested Changes or Clarifications
> > The authors state for the cylinder: "Since the domain is not rectangular, FNO cannot be directly applied without sacrificing the power FFTs." However, the direct method here seems to be to cut vertically and flatten the cylinder, giving a rectangular grid. And since a FFT assumes periodic boundary conditions, a FNO would be rotationally equivariant both in the angular direction (correctly) and in the vertical direction (incorrectly). Is this not also what is done by the CT-FNO in this case? Possibly I am misunderstanding the setup here.
> - Your understanding is correct. However, **for the vertical direction, padding will be applied to avoid (incorrect) equivariance in the vertical direction. This is discussed in Section 4.2 below equation (4).**
>
> > t is stated that the GNO can also be used such that it is equivariant - then this should be done and added to the comparison.
> - Following your suggestions, **we have added the comparison**. However, it is worth noting that GNO performs worse than FNO for normal testing data, so as expected, equivariant-GNO cannot outperform CT-FNO. We included this comparison in Table 1 and included more discussions about equivariant GNO and how the equivariance is achieved in Appendix C.2.
>
> > It was unclear to me why the performance on the rotated test set should be worse than on the normal test set for the comparison methods. Are the initial conditions of the training set not generated uniformly across angles?
> - **The initial conditions are not uniform across angles.**
> - For the Darcy Flow equation, we use the dataset directly from [1]. In this dataset, the coefficient functions are sampled from Gaussian Random Fields (GRFs) and then mapped to piecewise constants. As rotating GRFs does not change their distribution, the performance of the non-equivariant FNO does not degrade significantly in Sec. 5.2 (Darcy Flow).
> - To demonstrate the effectiveness and importance of the equivariant nature of CT-FNO, we chose an example in which the initial conditions are sampled in a way that rotations will change their distribution (in other words, they are not generated uniformly across angles) by including higher initial heat at fixed locations. **In many real-world data distributions, rotations will indeed change the distribution.** We have provided more details on how the initial conditions are generated in Appendix C.1. Additionally, we fixed an issue in the math expression in C.1 (missing the second half of the equation).
>
> > How are the vertical boundaries of the grid in CT-FNO treated compared to the angular (periodic) boundary?
> - As mentioned in the previous question, we apply padding to the radial axis to avoid equivariance issues on the radial axis. This, in a way, also addresses your concern here. **By applying padding, the radial axis no longer exhibits "translational" symmetries, and the periodicity that comes from convolution is also removed.**

---

> ### Author Response · Authors · 2024-05-29
> **Reply to Reviewer mw3p by Authors (3rd Part)**
>
> ### Regarding Optional and Minor Changes
> > ### Vague abstract
> - Thank you so much for the constructive feedback; **we have made the necessary changes**. Regarding your last point, "a single model could be trained on one domain shape and then generalize at test time to a new domain shape," this is a very interesting perspective, and we will explore it further in future work.
>
> > ### Missing general definition of CT-FNO
> - Thank you so much for the feedback. **We have included a general statement in Sec. 4.**
>
> >### Sampling grid and Fig.4
>
> > Fig. 4 was visually confusing: panel a) corresponds to d), and panel b) to c) ?
> - Sorry for this inconsistency. There was a typo in the caption. **Panel a) corresponds to c), and panel b) corresponds to c). We have corrected this.**
>
> > why not represent this as a rectangle with aspect ratio $1 : 2 \pi$?
> - Heuristically, we have to set up a sampling grid, which can be made $1 : 2 \pi$, and then just apply the floor or ceiling operator after sampling. As the difference is negligible, for simplicity, in this work, we consider $1:6$. **We have included such descriptions in the paper.**
>
> > Could the authors discuss when this is a useful or problematic feature?
> - **The message we aim to deliver is that the sampling grid must be carefully designed based on the specific application and evaluation criteria.** We did not include a detailed discussion on specific examples in the paper, as the optimal design of the sampling grid is highly application-dependent and should be decided for each case individually.
> - In certain scenarios, such as in rotating physical systems, non-uniform resolution can either be advantageous or problematic. For example, in these systems, the characteristic angular size of feature fluctuations can vary along the radial coordinate. Non-uniform resolution can be useful if it matches the varying scales of features across the radial axis, providing higher resolution where feature changes are more significant. Conversely, it can be problematic if the non-uniform resolution leads to undersampling or oversampling in regions where it is not appropriate, potentially missing critical details or creating redundant data. **Thus, understanding the specific requirements of the application is crucial in designing an effective sampling grid. **
> - We chose not to include this discussion to avoid lengthening the paper, but **we have included a general description in Section 4.3.**
>
> > ### Omitted proof and unclear significance of Corollary 4.1
> - Thank you so much for your suggestions. **The main aim of this is to justify why we can consider a different coordinate system for neural operators to learn.** We emphasize this in the main text to ensure that the readers can see this point, as a coordinate transformation to the PDE does not essentially change the PDE. As much as we would like to formalize it, the idea is relatively simple; thus, it is rather a straightforward extension from Kovachki et al. '21. We could elaborate more on this, but it would essentially be a restatement from Kovachki et al. '21, so to **give them credit, we prefer to direct the readers to their paper directly.**
>
> >### Discussion of discovering symmetries
> - Thank you so much for your comment. **This is a great point, and this could be one of our future works.** Recently, there have been papers published on automatically discovering symmetries, such as [2]. Coordinate transformation can be of potential in this venue, although it is not in the scope of this work. **We did not include more discussions on this because it is too early to conclude anything about our method in this direction.**
>
> >### Typos
> - Thank you so much for your comment. **We have fixed them.**

---

> > ### Comment · Reviewer_mw3p · 2024-06-02
> > **Thanks for the revision**
> >
> > Thank you for your work on the revision and response. My concerns have largely been addressed by the changes to the text and the additional experiment.
> > Concerning the distribution of initial conditions for the heat diffusion on the cylinder, thanks for the clarification, I would recommend to mention in sect. 5.1 or in Table 1 that the rotated test set is out-of-distribution. This may make the numbers in the table more easily interpretable for readers without looking into the appendix.

---

> > > ### Author Response · Authors · 2024-06-03
> > > **Reply to comment**
> > >
> > > Yes, it should be mentioned in the main text about the change in distribution after rotations, which will make the rotation test easier to understand for readers. **We have updated our manuscript based on your suggestion in Sec. 5.1 under Setup to include such a description.** We would like to, again, thank you for your time and efforts in reviewing our work and providing extremely valuable suggestions, which significantly improve the quality of our work. If you have any further concerns or comments, please do not hesitate to ask; we, as always, will try our best to address any concern you have.

---

### Review · Reviewer_5vdP · 2024-05-18

**Summary Of Contributions:**

This paper proposes the Coordinate Transform Fourier Neural Operator (CT-FNO) framework, which extends applicability of the Fourier Neural Operator (FNO) to non-rectangular domains, while preserving underlying symmetry. The paper also provides an extension of an FNO universal approximation theorem to CT-FNOs. Further, the paper conducts experiments that showcase how CT-FNOs are able to effectively maintain symmetries and how they perform as well if not better than a number of pertinent baselines in a variety of contexts. These contexts are learning the diffusion of heat on a cylinder, learning a synthetic operator, and learning the dynamics of a 2D Darcy flow equation.

**Audience:**

Yes

**Broader Impact Concerns:**

This work deals with broadening the applicability of Fourier Neural Operators. I do not believe the paper requires a statement addressing potential ethical implications of the presented research.

**Claims And Evidence:**

Yes

**Requested Changes:**

The writing in this paper is for the most part clear, however English conventions need to be followed more carefully in certain paragraphs. I elaborate on this further with a non-exhaustive list of corrections below. Additionally, the way standard deviations are given for tables is non-standard. Please use either $\pm$ or explicitly specify that standard deviations are given in parentheses next to each result. Also, what are "accuracies" in this context? I assume the authors are referring to relative error of the predicted functions, e.g. mean absolute percentage error (MAPE). Please augment the experimental section to clearly specify the metrics used for the presented results.

The aforementioned non-exhaustive list of minor corrections is given below:

- Section 2.1, first paragraph: "Fourier neural operators (FNOs) has drawn lots of attention" -> "Fourier neural operators (FNOs) have drawn lots of attention"
- Section 2.3, first paragraph: "For PDEs, the symmetries it can have" -> "For PDEs, the symmetries they can have"
- Section 3.1, second paragraph: "$L^\mathcal{X}_g$ and $L^\mathcal{Y}_g$ can be considered as a transformation, such as rotation, of the input" -> "$L^\mathcal{X}_g$ and $L^\mathcal{Y}_g$ can be considered as transformations, such as rotations, of the input"
- Section 3.2, first paragraph: "Thus, he integral kernel" -> "Thus, the integral kernel"
- Section 4.1, first paragraph: "In this works, we discuss" -> "In this work, we discuss"
- Section 4.2, third paragraph: "for Abelian Lie-groups" -> "for abelian Lie-groups" (the convention is to not capitalize the "a" in abelian)
- Figure 4, caption: The descriptions for (a) and (b) are flipped. Figure (a) is what happens when you use a 1:1 ratio, whereas Figure (b) is what happens when you use the ratio 1:6.

**Strengths And Weaknesses:**

## Strengths

1. The presented results are reasonable and showcase how CT-FNOs can learn over more complex domains without sacrificing accuracy (relative error) while preserving symmetry.

## Weaknesses

1. Although the idea of using a general coordinate transform to extend the applicability of CT-FNOs is a relatively new extension in the Fourier Neural Operator Space, the novelty is limited by a number of existing papers that explore closely related ideas. Perhaps the most related paper is Bonev et al. (2023), which uses a different coordinate system (spherical) for climate modeling via a Spherical Fourier Neural Operator (SFNO) extension of the FNO.

2. The paper does not present analysis of application areas that are now accessible via their presented coordinate transform augmentation and the demonstrated applicability is actually quite narrow; all non-rectangular domains are circular or effectively circular (e.g. the cylinder is a product of a circle and a unit interval). The same 1:6 sampling grid is reused, a study of other domain shapes is not given. To the credit of the authors, they are fairly upfront about this fact and state that broadening the applicability of the FNO is left for future work.

## Verdict

Although the presented idea of using a general coordinate transform to extend CT-FNOs is somewhat novel, closely related papers have investigated the use of a spherical coordinate transform to extend the applicability of FNOs. The application areas given are fairly synthetic and no examples are given beyond applications to a circular (or effectively circular) domain. That being said, the presented results for all three examples are reasonable and showcase how CT-FNOs can learn over more complex domains without sacrificing accuracy while preserving symmetry. I believe this paper is a reasonable first step in extending FNOs to work over more complex domains. The claims made in the submission are reasonably well-supported, although some of the language regarding broad applicability should be toned down to reflect the relatively limited scope of studied domains. I do believe the findings of the paper would be interesting to some portion of TMLR's audience, specifically, those working on FNOs. Since both criteria for acceptance to TMLR are satisfied, I recommend acceptance of this paper.

---

> ### Author Response · Authors · 2024-05-29
> **Reply to Reviewer 5vdP by Authors**
>
> Thank you for reviewing our work; we greatly value your insights and feedback, and thank you for your positive verdict and for accepting our work.
> ### Regarding Weakness No.1:
> - Yes, we admit that the idea of coordinate transformation is relatively simple; **while the idea itself is simple, combining coordinate transformations with symmetries for PDEs is relatively more sophisticated and is the main focus of this work**. In traditional numerical methods, it is very common to consider coordinate transformations. Bonev et al. (2023) is indeed closely related to this work; we provide more of a generalization of using coordinate transformations for FNO.
>
> ### Regarding Weakness No.2:
> - Yes, most domains are effectively circular, and **we believe that a generalization to other domain shapes has high potential. We will investigate this further as future work**.
>
> ### Regarding Requested Changes:
> - Thank you so much for catching the errors. **We have corrected all of them. Additionally, based on your concerns, we reviewed the paper several times to further improve the quality of our writing.**

---

> > ### Comment · Reviewer_5vdP · 2024-06-18
> > **Thank you for the updates**
> >
> > I have read your response and read over the changes in the revised paper. Thank you for making the necessary corrections.

---

### Review · Reviewer_pdUD · 2024-05-27

**Summary Of Contributions:**

The present paper introduces coordinate transform into Fourier neural operator together to extend the neural operator into many domains.

**Audience:**

Yes

**Broader Impact Concerns:**

No.

**Claims And Evidence:**

Yes

**Requested Changes:**

I don't have a request for major revisions, but only some minor changes.

- The text above Eq. (1): "Thus, he" should be "Thus, the"
- Fig. 3: it is better to add the labels of the x and y axes.
- The last paragraph in section 4: it is better to use the Jacobian matrix in the change of integration variables to strengthen the math rigor of why the radius and angular axes have a ratio of 1:6.

**Strengths And Weaknesses:**

I like the idea of using coordinate transform (canonical coordinate) to simplify the network implementation of modeling PDF and operators. That is the symmetry along an arbitrary manifold can be converted into simple translation symmetry by using the canonical coordinate. I believe this is an important extension of the Fourier neural operator.

---

> ### Author Response · Authors · 2024-05-29
> **Reply to Reviewer pdUD by Authors**
>
> Thank you so much for your comments and thank you for taking the time to review our work; your insights and feedback are greatly appreciated.
>
> **We have made all the necessary changes based on your feedback.** We fixed the typo and added the labels. For the last requested change, it is more of a heuristic based on the application to choose an appropriate sampling grid. We have added a general description in the paper explaining why this sampling ratio was chosen in the second paragraph of Sec. 4.3.

---

> > ### Comment · Reviewer_pdUD · 2024-06-10
> > **Thanks for the response**
> >
> > Thanks for the authors' reply and the revision of the manuscript. I like that the revised manuscript clearly identifies the newly added text. They address all my concerns. Overall, I like the idea of utilizing coordinate transform to simplify the implementation of neural operators.

---

### Author Response · Authors · 2024-05-29
**Reply to Reviews by Authors**

We would like to thank all the reviewers for their constructive feedback, which has greatly improved our paper. We have revised the manuscript and added new experiments, to address the reviewers' concerns. All changes in the manuscript are marked in red. Point-to-point responses are provided below. We also extend our gratitude to the Action Editor for their support and efforts in overseeing the review process.

---

### Decision · Action_Editor_ePdM · 2024-09-13

**Recommendation:** Accept as is

**Comment:**

This paper proposes an extension of Fourier Neural Operators to non-rectangular domains via general coordinate transformations. The resulting method preserves underlying symmetries and is applied in several physical modeling experiments.

The reviewers found the method to be a useful if relatively small improvement over prior work. They also found the scope of the analysis to be somewhat narrow. To enhance the impact of this work, the authors might consider extensions to more diverse and complex domains. Nevertheless, as it stands, the paper is well presented, with its main claims being adequately investigated and its main limitations being clearly described. As such, this paper is appropriate for publication at TMLR and I recommend acceptance.

**Audience:**

Yes, some members of the community will be interested in this paper.

**Claims And Evidence:**

Yes, the claims in this paper are supported by convincing evidence.